# Role of Systemic Immune Inflammation Index, Systemic Immune Response Index, Neutrophil Lymphocyte Ratio and Platelet Lymphocyte Ratio in Predicting Peritoneal Culture Positivity and Prognosis in Cases of Spontaneous Bacterial Peritonitis Admitted to the Emergency Department

**DOI:** 10.3390/medicina60081335

**Published:** 2024-08-16

**Authors:** Mehmet Göktuğ Efgan, Hüseyin Acar, Efe Kanter, Süleyman Kırık, Tutku Duman Şahan

**Affiliations:** Department of Emergency Medicine, Izmir Katip Çelebi University, 35620 Izmir, Turkey; goktugefgan@gmail.com (M.G.E.); dracar@hotmail.com (H.A.); efekanter@hotmail.com (E.K.); kiriksuleyman2107@outlook.com (S.K.)

**Keywords:** systemic inflammatory parameters, spontaneous bacterial peritonitis, emergency department

## Abstract

*Background and Objectives*: Spontaneous bacterial peritonitis (SBP) is a life-threatening disease that requires early diagnosis and treatment. It is known that a positive culture result for SBP, which is a common reason for admission to the emergency department, is related to the severity and prognosis of the disease. However, as it is not possible to determine the culture result in the early stage of the disease, different methods are required to predict prognosis in the emergency department. This study was conducted to evaluate the success of the SII, SIRI, NLR and PLR in predicting culture results, intensive care needs and mortality in patients with SBP admitted to the emergency department. *Materials and Methods*: This study was a retrospective, observational study. Patients with SBP who applied to the emergency department were included in this study. Pregnant women, patients with a malignancy, patients with another infection and patients with liver failure were excluded from this study. Data were analyzed in terms of culture results, the need for intensive care and mortality development. Analyses were performed using SPSS version 26. Results are presented with a 95% confidence interval. A *p* value less than 0.05 was considered statistically significant. Participant data were analyzed using the independent samples *t*-test or the Mann–Whitney U test based on normality, and ROC analyses were conducted to assess test accuracies and determine cut-off values. *Results*: A total of 275 patients were included in this study. Although the culture results of 183 patients were positive, 92 were negative. The SII, NLR and PLR were found to be significantly higher in culture-positive patients (*p* < 0.001, *p* = 0.013 and *p* = 0.002, respectively). The SII and NLR were found to be significantly higher in patients with high mortality (*p* < 0.001 and *p* = 0.017, respectively). *Conclusions*: This study showed that the SII, NLR and PLR may be useful in predicting culture positivity and prognosis in SBP patients in the emergency department.

## 1. Introduction

Spontaneous bacterial peritonitis (SBP) is a serious infection that develops in the presence of cirrhosis and ascites, and has a high risk of mortality. SBP is a disease frequently encountered in the emergency department, and its risk of mortality can be reduced when diagnosed and treated early [1]. For the diagnosis of SBP, the presence of at least 250/mm^3^ of neutrophils in a cell count made from the ascitic fluid sample is diagnostic, and culture positivity is not observed in every case. However, studies show that culture positivity is associated with both high inflammation levels and poor prognosis in SBP cases [2,3,4]. If the culture positivity of patients diagnosed with SBP in the emergency department could be known in advance, more aggressive treatment could be given in the early stage, and perhaps the prognosis could be improved. It is not possible for these patients’ culture tests to be completed during their stay in the emergency department. However, it may be possible to predict culture positivity by evaluating the high inflammatory response.

The systemic immune–inflammation index (SII), systemic inflammatory response index (SIRI), neutrophil-to-lymphocyte ratio (NLR) and platelet-to-lymphocyte ratio (PLR), calculated based on laboratory parameters, are tools that measure the severity of inflammation, and are used to predict the prognosis of many diseases, such as various cancers and tumors, infections, ischemic stroke, rheumatic diseases and pulmonary embolism [5,6,7,8,9]. It may be possible that these scores, which are indicators of poor prognostic outcome based on the severity of inflammation, can be used as diagnostic tools for cases of suspected peritonitis in the emergency department and as prognostic indicators in cases of diagnosed peritonitis.

The aim of this study was to evaluate the ability of the SII, SIRI, NLR and PLR to predict culture positivity and prognosis in patients diagnosed with SBP in the emergency department.

## 2. Materials and Methods

### 2.1. Study Design

This study was designed as a retrospective observational study. This study includes patients who applied to the emergency department between 1 March 2018 and 1 August 2023. Local ethics committee approval was obtained before starting this study.

### 2.2. Study Population

This study included patients aged 18 and over who applied to the emergency department and were diagnosed with peritonitis. Among these patients, we excluded those with additional pathologies that could cause abdominal pain, additional diagnoses of infection, pregnant or breastfeeding women, those with missing data, those with a history of malignancy and those diagnosed with liver failure. Additionally, patients who were referred to an external center and whose outcome information could not be obtained were also excluded from this study.

### 2.3. Data Collection

In order to determine which patients to include in this study, patient records were accessed through the hospital information management system. To identify patients diagnosed with peritonitis in the emergency department within the specified date range, a search was made in the hospital’s information management system using ICD10 diagnosis codes “K65, K65.0, K65.8 and K65.9” and the results were scanned in patient files and their diagnoses were confirmed. A total of 365 patients were identified. Of these patients, 39 were excluded due to a history of malignancy, 31 due to missing data, 15 due to additional infections, 3 due to pregnancy and 2 patients due to liver failure. The remaining 275 patients were included in this study. A flow chart of this study is given in Figure 1. Age, gender, laboratory data, imaging data of all patients, hospitalization times, mortality and comorbidities were recorded in the data record form for use in statistical analysis. Patients were divided into two groups according to whether the peritoneal culture result was positive or negative. The usability of the calculated SII, SIRI, NLR and PLR values in distinguishing between the two groups was investigated. Additionally, whether SII, SIRI, NLR and PLR values could be used as prognostic indicators in all patients diagnosed with peritonitis was examined.

### 2.4. Outcomes

The primary outcome of this study was that SII, SIRI, NLR and PLR values can predict the presence of growth in peritoneal culture in patients diagnosed with peritonitis. The secondary outcome was to evaluate whether SII, SIRI, NLR and PLR values can be used as prognostic indicators by predicting mortality and the need for intensive care in patients diagnosed with peritonitis.

### 2.5. Measurement Tools

In this study, calculations were made using the hemogram results obtained for each case. SII, SIRI, NLR and PLR values were calculated using the following formulas

-SII: (N × P)/L-SIRI: (N × M)/L-NLR: N/L-PLR: P/L

where N represents neutrophil, P represents platelet, L represents lymphocyte and M represents monocyte values.

### 2.6. Statistical Analysis

SPSS version 26 for Windows software (IBM Corporation, Armonk, NY, USA) was used for all analyses. For *p*-values, values less than 0.05 were considered significant. All statistics were calculated with a 95% confidence interval. Descriptive statistics were presented with frequency, percentage, mean, standard deviation, median and minimum and maximum values. Normality assumptions were tested with the Shapiro–Wilk test, skewness–kurtosis values and QQ plots. Data of participants were compared with the independent samples *t*-test if they fit the normal distribution, and with the Mann–Whitney U test if they did not. ROC analyses were performed to evaluate the success of the tests and determine cut-off values. From the coordinates of the ROC curve table, the values at which sensitivity and specificity were highest together were determined as the cut-offs.

## 3. Results

A total of 275 patients were included in this study, and 122 of them were women. The average age was calculated as 50.72 ± 29.28. The peritoneal culture was found to be positive in 92 (33.5%) of these patients. After evaluation in the emergency department, 57 (20.7%) of these patients were admitted to the intensive care unit. In total, in-hospital mortality occurred in 119 (43.3%) patients. Descriptive statistics of these patients are presented in Table 1.

ROC curve analyses were performed for the ability of SII, SIRI, NLR and PLR variables to predict peritoneal culture results, and were found to be statistically significant. The cut-off point for the SII value was found to be >1228.45, and the area under the curve was 0.633. For this value, sensitivity was calculated as 65.20% and specificity as 53.00%. The cut-off point for the NLR value was found to be >6.00, and the area under the curve was 0.592. For this value, sensitivity was calculated as 57.60% and specificity as 55.20%. The cut-off point for the PLR value was found to be >176.17, and the area under the curve was 0.614. For this value, sensitivity was calculated as 64.10% and specificity as 55.20%. Cut-off scores, AUC values, sensitivity, selectivity values and ROC curve analyses for SII, SIRI, NLR and PLR variables for predicting growth in peritoneal cultures are presented in Table 2 and Figure 2.

A comparison between groups with positive and negative peritoneal culture results is presented in Table 3. There were statistically significant differences between the two groups for SII, NLR, PLR, neutrophil and platelet variables. The SII was calculated as 2718.21 ± 2504.32 for the culture-positive group and 2229.05 ± 3507.26 for the culture-negative group. The SII value was higher for the culture-positive group, and this value was statistically significant. The NLR was calculated as 9.72 ± 7.6 for the culture-positive group and 9.49 ± 16.33 for the culture-negative group. The NLR value was higher in the culture-positive group, and this value was statistically significant. The PLR was calculated as 422.91 ± 982.96 for the culture-positive group and 251.31 ± 260.03 for the culture-negative group. The PLR value was higher for the culture-positive group, and this value was statistically significant. Neutrophils were calculated as 9.45 ± 6.63 for the culture-positive group and 8.84 ± 9.69 for the culture-negative group. The neutrophil value was higher in the culture-positive group, and this value was statistically significant. The platelet count was calculated as 285.11 ± 142.24 for the culture-positive group and 238.87 ± 140.87 for the culture-negative group. The platelet value was higher for the culture-positive group, and this value was statistically significant (Table 3).

ROC curve analyses were performed regarding the use of SII, SIRI, NLR and PLR variables to predict the need for intensive care, which were found to be statistically significant. The cut-off point for the SII value was found to be >1236.70, and the area under the curve was 0.605. For this value, sensitivity was calculated as 64.90% and specificity as 51.40%. The cut-off point for the NLR value was found to be >5.88, and the area under the curve was 0.600. For this value, sensitivity was calculated as 61.40% and specificity as 52.30%. The cut-off point for the PLR value was found to be >193.84, and the area under the curve was 0.599. For this value, sensitivity was calculated as 63.20% and specificity as 58.30%. Cut-off scores, AUC values, sensitivity, selectivity values and ROC curve analyses of SII, SIRI, NLR and PLR variables for predicting the need for intensive care are presented in Table 4 and Figure 3.

ROC curve analyses were performed regarding the use of SII and NLR values to predict mortality, which were found to be statistically significant. The cut-off point for the SII value was found to be >1296.63, and the area under the curve was 0.312. For this value, sensitivity was calculated as 63.00% and specificity as 62.20%. The cut-off point for the NLR value was found to be >5.92, and the area under the curve was 0.584. For this value, sensitivity was calculated as 58.80% and specificity as 55.80%. Cut-off scores, AUC values, sensitivity, selectivity values and ROC curve analyses of SII, SIRI, NLR and PLR variables for predicting mortality are presented in Table 5 and Figure 4.

## 4. Discussion

This study evaluated the success of the SII, SIRI, NLR and PLR in predicting culture positivity and prognosis in patients diagnosed with SBP in the emergency department, and showed that the SII, NLR and PLR, but not the SIRI, could predict culture positivity and the need for intensive care, and that the SII and NLR could predict mortality.

To our knowledge, this study is the first in the literature to evaluate the success of indexes measuring the severity of inflammation, such as the SII, SIRI, NLR and PLR, in predicting culture positivity in SBP. However, a study by Sun Hee Na et al. found a statistically significant increase in the number of neutrophils in ascitic fluid in culture-positive peritonitis compared to culture-negative ones. However, in the same study, neutrophil numbers in peripheral blood were found to be similar in both groups. This study did not include any data on lymphocyte, monocyte and platelet counts [2]. In our study, the SII, NLR and PLR, calculated using neutrophil, lymphocyte and platelet counts, were found to be successful in predicting peritoneal culture results, while the SIRI, the only index that included monocytes, was found unsuccessful. It is known that, in addition to the number of neutrophils increasing in the early stage when inflammation develops, platelets also increase due to inflammation [10,11]. Therefore, it is possible that scores using neutrophils and platelets, such as the SII, NLR and PLR, indicate culture positivity due to increased inflammation in SBP. However, the fact that monocytes increase mostly in the late stage of inflammation [12] may be the reason why the SIRI index could not predict culture positivity when examined in the emergency department. Comparing the indices examined in the early and late stages may be informative in this regard.

When we reviewed the literature, the success of the NLR, PLR, SII and SIRI in predicting prognosis in patients with SBP was an issue that had not yet been studied. In a study conducted by Yan Yang et al., it was observed that the SII and SIRI were successful in predicting mortality in patients receiving peritoneal dialysis, even if peritonitis was not present [13]. Zhou et al., in another study, reported that the NLR, PLR and SII were associated with treatment failure in patients receiving peritoneal dialysis. In our study, while the SII, NLR and PLR were found to be successful in predicting the need for intensive care in patients with SBP, the SIRI was found unsuccessful. Additionally, while the SII and NLR were found to be associated with mortality, the SIRI and PLR were unrelated. Thrombocytes play a role in the resolution of inflammation as well as in its development [14]. This may be the reason why the PLR rate could not predict the need for intensive care and mortality in our study. Additionally, research has shown that platelets act as immunomodulators by interacting with monocytes during the inflammation process [15,16,17]. Perhaps this is why the SIRI, which includes the monocyte count in the calculation, seemed to fail in predicting mortality in patients with SBP. Further research is needed on this subject.

This study had some limitations. The primary limitation was its retrospective, single-center design, which, along with the small population size, may have negatively affected the results. Additionally, diagnosing peritonitis posed challenges due to the lack of clear criteria, and diagnoses were based on the internal medicine team’s assessment. Future studies should consider a prospective design with more stringent diagnostic criteria to address these issues.

## 5. Conclusions

This study showed that the SII, NLR and PLR may be useful in predicting culture positivity and prognosis in SBP patients in the emergency department. However, more studies are needed on this subject.

## Figures and Tables

**Figure 1 medicina-60-01335-f001:**
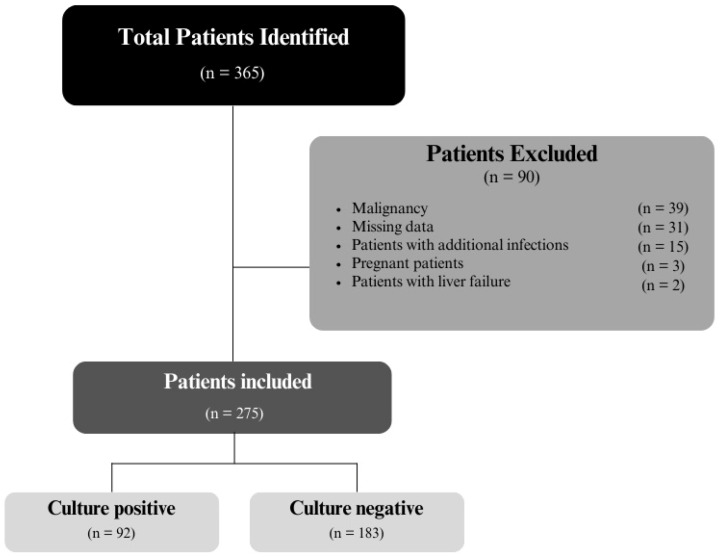
Flow chart of this study.

**Figure 2 medicina-60-01335-f002:**
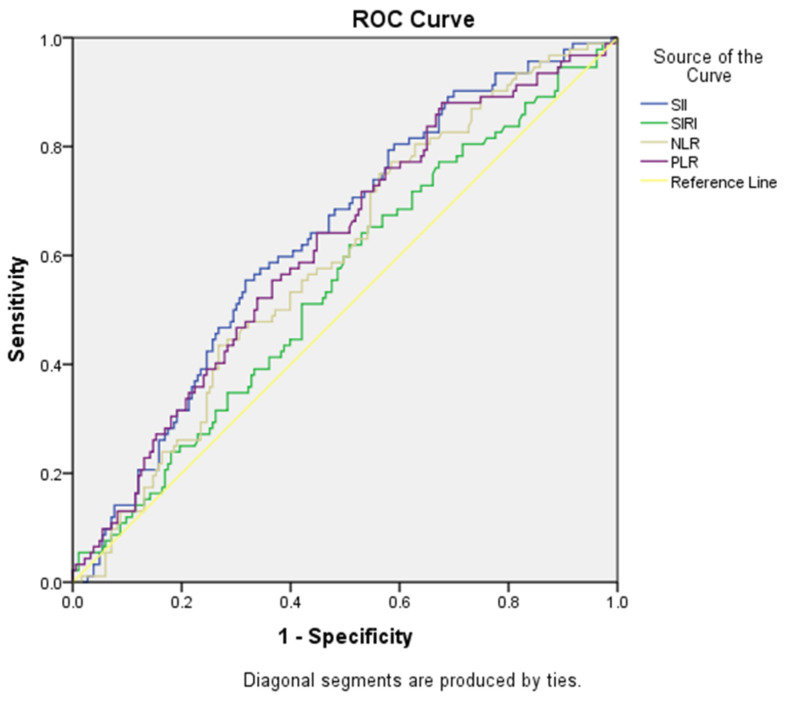
ROC curves evaluating the success of SII, SIRI, NLR and PLR variables in predicting peritoneal culture outcomes.

**Figure 3 medicina-60-01335-f003:**
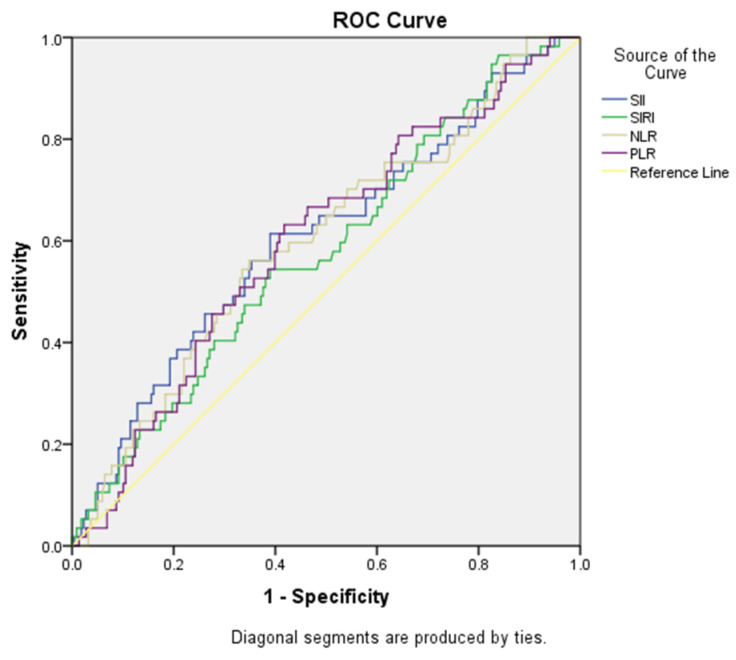
ROC curves for SII, SIRI, NLR and PLR variables in predicting the need for intensive care.

**Figure 4 medicina-60-01335-f004:**
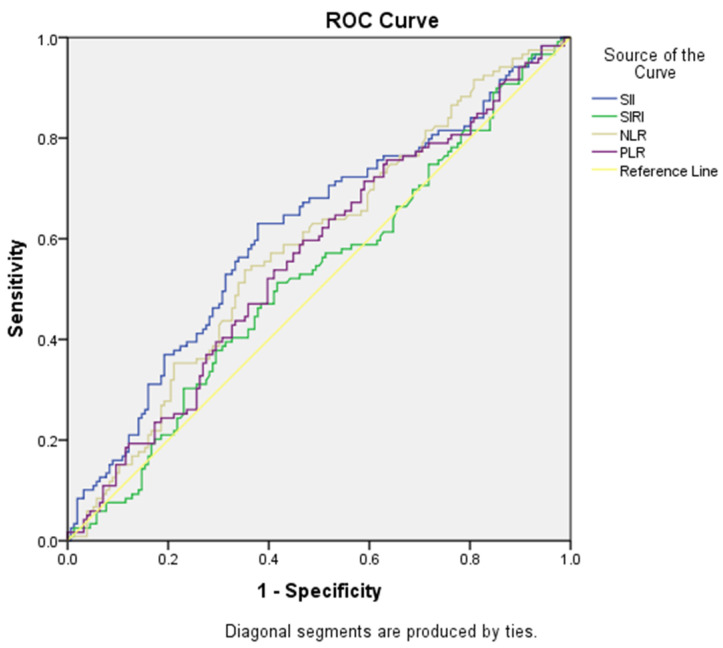
ROC curves for SII, SIRI, NLR and PLR variables in predicting mortality.

**Table 1 medicina-60-01335-t001:** Descriptive characteristics.

	Statistics
**Age**	50.72 ± 29.28
**Gender**	
Female	122 (44.4%)
Male	153 (55.6%)
**Spontaneous or secondary peritonitis**	
Spontaneous	156 (56.7%)
Secondary	119 (43.3%)
**Peritoneal culture result**	
Negative	183 (66.5%)
Positive	92 (33.5%)
**Need for intensive care**	
No	218 (79.3%)
Yes	57 (20.7%)
**Exitus**	
No	156 (56.7%)
Yes	119 (43.3%)
**Number of days in hospital**	14.42 ± 11.29
**CRP**	75.69 ± 84.88
**Albumin**	27.30 ± 10.79
**Leukocyte**	11.73 ± 7.45
**Neutrophil**	9.05 ± 8.78
**Lymphocyte**	2.16 ± 8.01
**Platelet**	254.34 ± 142.76
**Monocyte**	0.86 ± 0.57
**RDW**	16.55 ± 3.30
**INR**	1.42 ± 0.46
**SII**	2392.70 ± 3210.46
**SIRI**	7.66 ± 9.85
**NLR**	9.57 ± 14.01
**PLR**	308.72 ± 610.23

CRP: C-reactive protein, RDW: red cell distribution width, INR: international normalized ratio, SII: systemic immune–inflammation index, SIRI: systemic inflammatory response index, NLR: neutrophil-to-lymphocyte ratio, PLR: platelet-to-lymphocyte ratio.

**Table 2 medicina-60-01335-t002:** Cut-off values, AUC values, sensitivity, specificity and statistical significance of SII, SIRI, NLR and PLR variables for predicting peritoneal culture results.

Test Result Variables	Cut-Off	AUC	Std. Error	*p*	Asymptotic 95% Confidence Interval		Sensitivity	Specificity
					Lower Bound	Upper Bound		
**SII**	**>1228.45**	**0.633**	**0.034**	**<0.001**	**0.565**	0.700	65.20	53.00
SIRI	>4.29	0.546	0.037	0.209	0.475	0.618	53.30	53.00
NLR	>6.00	0.592	0.035	**0.013**	0.523	0.661	57.60	55.20
PLR	>176.17	0.614	0.035	**0.002**	0.545	0.684	64.10	55.20

SII: systemic immune–inflammation index, SIRI: systemic inflammatory response index, NLR: neutrophil-to-lymphocyte ratio, PLR: platelet-to-lymphocyte ratio.

**Table 3 medicina-60-01335-t003:** Comparison of all variables between groups with positive and negative peritoneal cultures.

	Peritoneal Culture Results		Test Statistic	*p*
	Positive	Negative		
**Age**	56.28 ± 25.22	47.93 ± 30.80	−1.690	0.091
**SII**	2718.21 ± 2504.32	2229.05 ± 3507.26	−3.598	**<0.001**
**SIRI**	8.80 ± 12.24	7.08 ± 8.38	−1.256	0.209
**NLR**	9.72 ± 7.61	9.49 ± 16.33	−2.497	**0.013**
**PLR**	422.91 ± 982.96	251.31 ± 260.03	−3.096	**0.002**
**WBC**	13.04 ± 8.86	11.07 ± 6.55	−1.789	0.074
**Neutrophil**	9.45 ± 6.63	8.84 ± 9.69	−2.051	**0.040**
**Lymphocyte**	2.12 ± 8.36	2.18 ± 7.85	−1.144	0.253
**Platelet**	285.11 ± 142.24	238.87 ± 140.87	−3.106	**0.002**
**Monocyte**	0.84 ± 0.57	0.86 ± 0.57	−0.074	0.941

SII: systemic immune–inflammation index, SIRI: systemic inflammatory response index, NLR: neutrophil-to-lymphocyte ratio, PLR: platelet-to-lymphocyte ratio.

**Table 4 medicina-60-01335-t004:** Cut-off values, AUC values, sensitivity, specificity and statistical significance of SII, SIRI, NLR and PLR variables for predicting the need for intensive care.

Test Result Variables	Cut-Off	AUC	Std. Error	*p*	Asymptotic 95% Confidence Interval		Sensitivity	Specificity
					Lower Bound	Upper Bound		
**SII**	**>1236.70**	**0.605**	**0.043**	**0.014**	**0.521**	0.690	64.90	51.40
SIRI	>4.17	0.577	0.042	0.074	0.495	0.659	56.10	51.40
NLR	>5.88	0.600	0.042	**0.020**	0.517	0.682	61.40	52.30
PLR	>193.84	0.599	0.041	**0.021**	0.519	0.680	63.20	58.30

SII: systemic immune–inflammation index, SIRI: systemic inflammatory response index, NLR: neutrophil-to-lymphocyte ratio, PLR: platelet-to-lymphocyte ratio.

**Table 5 medicina-60-01335-t005:** Cut-off values, AUC values, sensitivity, specificity and statistical significance of SII, SIRI, NLR and PLR variables for predicting mortality.

Test Result Variables	Cut-Off	AUC	Std. Error	*p*	Asymptotic 95% Confidence Interval		Sensitivity	Specificity
					Lower Bound	Upper Bound		
**SII**	**>1296.63**	**0.312**	**0.035**	**0.001**	**0.545**	0.680	63.00	62.20
SIRI	>3.97	0.522	0.035	0.528	0.453	0.591	54.60	50.00
NLR	>5.92	0.584	0.035	**0.017**	0.516	0.651	58.80	55.80
PLR	>167.13	0.561	0.035	0.081	0.493	0.630	60.50	50.00

SII: systemic immune–inflammation index, SIRI: systemic inflammatory response index, NLR: neutrophil-to-lymphocyte ratio, PLR: platelet-to-lymphocyte ratio.

## Data Availability

Data may be shared with an appropriate justification if requested.

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
