# Peer review of "Role of Systemic Immune Inflammation Index, Systemic Immune Response Index, Neutrophil Lymphocyte Ratio and Platelet Lymphocyte Ratio in Predicting Peritoneal Culture Positivity and Prognosis in Cases of Spontaneous Bacterial Peritonitis Admitted to the Emergency Department"

_medicina, 2024, doi:10.3390/medicina60081335_

Round 1

Reviewer 1 Report

Comments and Suggestions for Authors

I read your manuscript with great interest,” Role of SII, SIRI, NLR, and PLR in Predicting Peritoneal Culture Positivity and Prognosis in Cases of Spontaneous Bacterial Peritonitis Admitted to the Emergency Department,” and I find your research useful.

However, I have some suggestions:

1.       In the abstract, it is mentioned, “Results: A total of 275 patients were included in 25 the study. While the culture results of 183 patients were positive, 92 were negative.” similar in the material and method section, “The remaining 275 patients were included in the study.” Meanwhile in the results section, it is mentioned: “A total of 218 patients were included in the study, and 94 (43.12%) of them were female.”  I think it is important to clarify this difference and to be consistent in presenting your data.

In the introduction section, I suggest broadening the disscusion regarding the existent guidelines on SBP.

2.       I1.       t is recommended that you introduce a figure for your study population, including and excluded, as this may help improve the confusion created by the difference in the number of included patients, as mentioned above.

3.       Also, consistency is needed in presenting numerical data -some are with comma separators and others with dots, and please use the same format in decimals ( ex, 2 for all your data ) For example: “endpoints was >12.391, sensitivity was 100.0% and specificity was 74.3%. For the PLR variable, the optimum cut off value for predicting those 123 with outcome was > 266,666, sensitivity was 100.0% and specificity was 78.0%.” , “o be >1228.45 and the area under the curve was 0.633. For this value, sensitivity 201 was calculated as 65.20%”, but it is needed to check the entire manuscript.

4.       In Table 1, what is the meaning of Fire :” Fire 37.54±4.66”?

5.       There are English language issues “ statisticks”. ( table 1) “Excitus” ( table 1),

6.       Please specify what Test statistics means in Table 3. ( is there a correlation parameter, which one?)

7.       You need to specify the strengths and limitations of your study. One of the limitations is the one that you mentioned in the introduction section: “For the diagnosis of SBP, the presence of at least 250/mm3  neutrophils in the cell count made from the ascitic fluid sample is diagnostic, and culture positivity is not observed in every case”, and your cohort may also include negative PBS with negative cultures. This should be mentioned.

8.       In my opinion, you should mention in your manuscript the latest guidelines for diagnosing and managing SBP because the management includes the administration of wide-spectrum antibiotics until the culture results are available. Therefore, in this regard, in the discussion section, you should mention how the results of the cultures will change the medical management since starting empiric antibiotics before the culture is available in SPB is recommended.

9.       The references need to be rechecked:” 14. 80. Margraf, A., & Zarbock, A. (2019). Platelets in Inflammation and Resolution. The Journal of Immunology, 203, 2357 – 2367”

10.   The manuscript needs English editing revision.

Comments on the Quality of English Language

Moderate  English editing is needed.

Author Response

Comments 1: In the abstract, it is mentioned, “Results: A total of 275 patients were included in 25 the study. While the culture results of 183 patients were positive, 92 were negative.” similar in the material and method section, “The remaining 275 patients were included in the study.” Meanwhile in the results section, it is mentioned: “A total of 218 patients were included in the study, and 94 (43.12%) of them were female.”  I think it is important to clarify this difference and to be consistent in presenting your data.

Response 1: In the manuscript, there are two Results section. We realised that the first one is and old version of the study. Therefore we removed the first part and so made the revisions for your comments 1, 3, 4 and 6.

Comments 2. It is recommended that you introduce a figure for your study population, including and excluded, as this may help improve the confusion created by the difference in the number of included patients, as mentioned above.

Response 2: Flowchart is added, stated as Figure 1 in manuscript.

Comments 3. Also, consistency is needed in presenting numerical data -some are with comma separators and others with dots, and please use the same format in decimals ( ex, 2 for all your data ) For example: “endpoints was >12.391, sensitivity was 100.0% and specificity was 74.3%. For the PLR variable, the optimum cut off value for predicting those 123 with outcome was > 266,666, sensitivity was 100.0% and specificity was 78.0%.” , “o be >1228.45 and the area under the curve was 0.633. For this value, sensitivity 201 was calculated as 65.20%”, but it is needed to check the entire manuscript.

Response 3: Revision is made for this comment as stated in response 1.

Comments 4. In Table 1, what is the meaning of Fire :” Fire 37.54±4.66”?

Response 4: Revision is made for this comment as stated in response 1.

Comments 5. There are English language issues “ statisticks”. ( table 1) “Excitus” ( table 1),

Response 5: Corrections are made for these particular words, also the word “Plathelet” is corrected.

Comments 6. Please specify what Test statistics means in Table 3. ( is there a correlation parameter, which one?)

Response 6: Revision is made for this comment as stated in response 1.

Comments 7. You need to specify the strengths and limitations of your study. One of the limitations is the one that you mentioned in the introduction section: “For the diagnosis of SBP, the presence of at least 250/mm3  neutrophils in the cell count made from the ascitic fluid sample is diagnostic, and culture positivity is not observed in every case”, and your cohort may also include negative PBS with negative cultures. This should be mentioned.

Response 7: Thank you for pointing this out, we made the limitations section stronger and mentioned about the diagnosis of PBS.

Comments 8. In my opinion, you should mention in your manuscript the latest guidelines for diagnosing and managing SBP because the management includes the administration of wide-spectrum antibiotics until the culture results are available. Therefore, in this regard, in the discussion section, you should mention how the results of the cultures will change the medical management since starting empiric antibiotics before the culture is available in SPB is recommended.

Response 8: You are absolutely right about what you said, but we evaluated whether there would be a positive culture in our study. Although it is very important to organize the management of antibiotics according to the culture result, we do not have the information about the microorganisms grown in the culture and the antibiogram results and they are outside our study objectives. However, your valuable suggestions are very valuable to us as they can be different study subjects. Thank you for enlightening us on this issue.

Comments 9. The references need to be rechecked:” 14. 80. Margraf, A., & Zarbock, A. (2019). Platelets in Inflammation and Resolution. The Journal of Immunology, 203, 2357 – 2367”

Response 9: Correction is made.

Comments 10. The manuscript needs English editing revision.

Response 10: As you requested, we have reviewed the English language and made the necessary corrections.

Reviewer 2 Report

Comments and Suggestions for Authors

Thank you for the opportunity to review this work. 

-The abstract section should include the type of statistics, the result with each statistic parameter, and the p-value. Moreover, please recheck the abbreviation "BPD patient" and include the full term in the first place. Thus, the abstract should be revised.

-In the method section, the authors stated, "ROC analysis was performed to evaluate the success of the tests and determine cut-off values." Please explicitly clarify the method for determining the cut-off values.

-When comparing the ROC curves of different prediction tools, it is essential to use statistical methods to demonstrate significant differences among them. For example, it could be reported that "NLR (AUC 0.835, 95% CI 0.759 to 0.912) versus PLR (AUC 0.856, 95% CI 0.788 to 0.925) have statistically significant differences (p = __)." It is necessary to compare each pair of the essential prediction tools.

-Numerous tables and figures were disorganized and required reordering and revising within the main text.

-There were some errors in typos, such as "Plathelet-to-lymphocyte ratio."

Author Response

-The abstract section should include the type of statistics, the result with each statistic parameter, and the p-value. Moreover, please recheck the abbreviation "BPD patient" and include the full term in the first place. Thus, the abstract should be revised.

Thank you for your review. As a typo, BPD was written instead of SBP, correction was made.

-In the method section, the authors stated, "ROC analysis was performed to evaluate the success of the tests and determine cut-off values." Please explicitly clarify the method for determining the cut-off values.

We agree with you on this issue. In line with your suggestions, we have made the necessary explanation on how cut-off values are calculated.

-When comparing the ROC curves of different prediction tools, it is essential to use statistical methods to demonstrate significant differences among them. For example, it could be reported that "NLR (AUC 0.835, 95% CI 0.759 to 0.912) versus PLR (AUC 0.856, 95% CI 0.788 to 0.925) have statistically significant differences (p = __)." It is necessary to compare each pair of the essential prediction tools.

Thank you for your evaluation. However, in this study, we evaluated the success of the parameters we have in predicting culture positivity, mortality and intensive care admission rates. When patients were divided into groups according to culture positivity and negativity, whether mortality developed or not, the significant differences and the relevant p values ​​were given as you stated. However, in this study, we did not aim to compare these scorings with each other statistically.-Numerous tables and figures were disorganized and required reordering and revising within the main text.

We have made the necessary arrangements, taking your evaluation into account.

-There were some errors in typos, such as "Plathelet-to-lymphocyte ratio."

This and similar typos have been corrected.

Reviewer 3 Report

Comments and Suggestions for Authors

This retrospective study presented the usefulness of SBP biomarkers in emergency umit. It concluded that SII, NLR, and PLR are effective predictors of SBP prognosis. It is a fantastic study based on real-world data. 1. The title is OK for reflecting the aim of the article. 2. The abstract is OK but does not need statistical analysis methods. 3. The introduction is okay. 4. The methods are acceptable. However, the authors are requested to make a flow chart for this study. 5. The results are satisfactory. 6. The discussion is concise but well-written. 7. The figures and legends are somewhat complicated. They could unite the ROC curve and their p-value. 9. The number of references is small but acceptable.

1. What is the main question addressed by the research? The article has discussed whether SII, SIRI, NLR, and PLR could be biomarkers for SBP. 2. Do you consider the topic original or relevant in the field? Does it address a specific gap in the field? Yes. 3. What does it add to the subject area compared with other published material? This article proposes non-invasive methods for predicting the outcomes of SBP. 4. What specific improvements should the authors consider regarding the methodology? What further controls should be considered? The authors could compare the bacterial differences in these biomarkers. 5. Are the conclusions consistent with the evidence and arguments presented, and do they address the main question posed? Yes. 6. Are the references appropriate? The number of references is considerably small for an original article. 7. Please include any additional comments on the tables and figures. The authors may place the ROC curve above the table of the p-value.

Comments on the Quality of English Language

Minor English editing is required. 

Author Response

Dear Editor, first of all, thank you very much for your evaluation and valuable contributions. We have taken your corrections into consideration.

  1. The title is OK for reflecting the aim of the article.

Thank you for your evaluation.

  1. The abstract is OK but does not need statistical analysis methods.

We agree with you about this subject. However, other editors especially wanted us to state the method of the statistical analysis. Thus, we made some improvements in the abstract.

  1. The introduction is okay.

Thank you for your evaluation.

  1. The methods are acceptable. However, the authors are requested to make a flow chart for this study.

We agree with this and we added a flow chart named Figure 1 to the manuscript.

  1. The results are satisfactory.

Thank you for your evaluation.

  1. The discussion is concise but well-written.

Thank you for your evaluation.

  1. The figures and legends are somewhat complicated. They could unite the ROC curve and their p-value.

Thank you for your assessment. Since we have so much data, combining all of it in a single figure or table would cause serious confusion, so we preferred to do it this way. However, we will pay attention to this in our future work.

  1. The number of references is small but acceptable.

Thank you for your evaluation.

Round 2

Reviewer 1 Report

Comments and Suggestions for Authors

Dear authors,

I appreciate the improvement of your manuscript. Still there are some issues:

1.       The phrase is incomplete or unclear and there is English language polishing needed

” SII and NLR were found to be significantly higher in patients with mortality”

Conclusions:” This study showed that SII, NLR, and PLR 31 may be useful in predicting culture positivity and prognosis in SBP patients in the emergency departme”

2.       In the results you stated that ( in table and text)

SII 2392,70±3210,46

 SIRI 7,66±9,85

NLR 9,57±14,01

PLR 308,72±610,23

These values represent the mean with std dev.?

Does this mean that you had negative ratios ( or values) for SII, SIRI, NLR, and PLR?

Same for CRP? 75.69±84.88?

Could you explain what these values represent?

Comments on the Quality of English Language

There is English language polishing needed.

Author Response

Dear rewiever,

Thank you for your comment about this manuscript. We have carefully edited your suggestions. The language of the entire article has been reviewed and edited. Edited areas are highlighted in yellow and red (the last ones). Explanations regarding statistics are below.

Dear authors,

I appreciate the improvement of your manuscript. Still there are some issues:

Comment 1:   The phrase is incomplete or unclear and there is English language polishing needed

” SII and NLR were found to be significantly higher in patients with mortality”

Conclusions:” This study showed that SII, NLR, and PLR 31 may be useful in predicting culture positivity and prognosis in SBP patients in the emergency departme”

Response 1: These sentences have been edited. They have been re-inserted into the text. They are marked in red.

Comment 2:      In the results you stated that ( in table and text)

SII 2392,70±3210,46

 SIRI 7,66±9,85

NLR 9,57±14,01

PLR 308,72±610,23

These values represent the mean with std dev.?

Does this mean that you had negative ratios ( or values) for SII, SIRI, NLR, and PLR?

Same for CRP? 75.69±84.88?

Could you explain what these values represent?

Response 2: These values ​​represent the mean and standard deviation. The reason for these results is that the distances from the mean have a fairly large range of variation, but negative values ​​are not present and are not possible.

Reviewer 2 Report

Comments and Suggestions for Authors

Thank you to the authors for this manuscript. The manuscript deserves publication.

Author Response

Thank you for your precious comments.

Reviewer 3 Report

Comments and Suggestions for Authors

The authors responded to reviewers' comments appropriately and revisited them well. 

Comments on the Quality of English Language

Minor English editing is required. 

Author Response

Thank you for your precious comments.